# Building Up a Piperazine Ring from a Primary Amino Group via Catalytic Reductive Cyclization of Dioximes

**DOI:** 10.3390/ijms241411794

**Published:** 2023-07-22

**Authors:** Evgeny V. Pospelov, Alexey Yu. Sukhorukov

**Affiliations:** N. D. Zelinsky Institute of Organic Chemistry, Leninsky Prospect, 47, Moscow 119991, Russia

**Keywords:** piperazines, reductive cyclization, Michael addition, oximes, heterogeneous catalytic hydrogenation

## Abstract

Piperazine is one of the most frequently found scaffolds in small-molecule FDA-approved drugs. In this study, a general approach to the synthesis of piperazines bearing substituents at carbon and nitrogen atoms utilizing primary amines and nitrosoalkenes as synthons was developed. The method relies on sequential double Michael addition of nitrosoalkenes to amines to give bis(oximinoalkyl)amines, followed by stereoselective catalytic reductive cyclization of the oxime groups. The method that we developed allows a straightforward structural modification of bioactive molecules (e.g., α-amino acids) by the conversion of a primary amino group into a piperazine ring.

## 1. Introduction

Among saturated *N*-heterocycles, piperazine is the second most frequently found scaffold in small-molecule FDA-approved drugs [1]. The piperazine moiety is found in various pharmaceuticals, such as antipsychotic, anticancer, antidepressant, antihistamine, antianginal, anti-inflammatory, antiviral, and imaging agents [2,3]. Moreover, the piperazine ring is the key structural motif in several blockbuster drugs, including Gleevec (Imatinib) and Viagra (Sildenafil). Apart from medical use, piperazines are commonly applied as pesticides [4], ligands in catalysts [5], CO_2_-capturing materials [6], as building blocks in crystal design [7,8], and in polymer production [9,10].

Given the importance of piperazines, numerous synthetic approaches have been developed to access the broad chemical space of these heterocycles [11,12]. A common strategy relies on the functionalization of the parent molecule, which is easily accomplished by the addition of electrophiles at *N*-atoms and cross-coupling reactions [13,14,15,16]. Direct functionalization at *C*-atoms in piperazine is challenging (although some progress has been achieved in recent years [17,18]), and the synthesis of carbon-substituted piperazines is often accomplished via cyclization of the corresponding linear diamine precursors [11,12]. Alternative strategies employing the hydrogenation of pyrazines [19], [3+3]-type dimerization of aziridines [20], ring-opening reactions in DABCO derivatives [21], and ring expansion reactions in imidazolines [22] are limited by the specific structure of the starting materials. Overall, the synthesis of polysubstituted piperazines still remains a challenging problem.

For drug design purposes, the construction of piperazine ring **1** from a primary amino group is a useful synthetic strategy (Figure 1). Since a plethora of pharmaceuticals and natural products contain a primary amino group, this synthetic tool would generate their potentially potent piperazine modifications [23]. Several attempts were undertaken to develop this methodology using 1,5-dihaloamines, diethanolamines, and their derivatives as cyclizing agents (Figure 1). For example, Watanabe et al. [24] reported a hydrogen borrowing strategy to assemble piperazines from simple amines and *N*-substituted diethanolamines, yet the yields of the products were low in many cases. Huang and Li et al. [25] developed a one-pot protocol for the conversion of primary amines into piperazines by cyclization with tosylbis(2-(tosyloxy)ethyl)amine. However, these methods were applied only to the synthesis of *C*-unsubstituted piperazine derivatives, most likely due to a limited availability of the corresponding substituted diethanolamine derivatives and their reduced reactivity in these reactions. Also, harsh reaction conditions are required, under which complex substrates may not be tolerated. To develop a more general method, we suggested a conceptually different strategy (Figure 1), which involves the conversion of a primary amine into bis(oximinoalkyl)amine **2** through double Michael addition to nitrosoalkenes (**NSA**), followed by catalytic reductive cyclization of the dioxime unit to give a piperazine ring. In our recent work, we have shown that a related cyclization of dioximes of 1,5-diones provides an expedient route to piperidines [26].

## 2. Results and Discussion

The first step of the suggested sequence requires the double oximinoalkylation of a primary amine through a Michael reaction with nitrosoalkenes **NSA**. Due to the instability of **NSA**, suitable precursors are used to generate these species in situ [27,28,29]. In the previous studies by our group [26,30] and other researchers [31], it has been shown that silylated ene-nitrosoacetals **3** are convenient precursors of **NSA** providing high chemoselectivity when coupling with various types of nucleophiles, including amines [32,33,34]. Using this methodology, a series of symmetrically substituted dialdo- and diketooximes **2a–o** was prepared by the treatment of simple primary amines with 2.1 equiv. of the corresponding ene-nitrosoacetals **3a–c** (R^1^ = H, Me, Ph, respectively, Figure 2), followed by methanolysis of the resulting *O*-silyl ethers. In most cases, dioximes **2** were formed in high yields, except for sterically hindered amines (*tert*-butylamine and α-phenylethylamine leading to products **2b**,**k**) and α-amino acid esters (products **2g**,**h**). In the latter case, the lower efficiency is likely due to the self-dimerization of α-amino acid esters to diketopiperazines as a side process. Also, in the reaction with L-leucine ethyl ester, a noticeable amount of a mono-addition product was obtained. 

Typically, dioximes **2** were obtained and characterized as mixtures of *E*/*Z*-isomers (the isomeric ratio slightly changed with time and depended on the solvent). For dialdooximes **2a–h**, *E*,*E*- and *E*,*Z*-isomers were formed in comparable quantities, while for diketooximes, the **2i–n** *E*,*E*-isomer was often predominant. The stereochemical assignment of oxime groups was performed on the basis of known relationships between the configuration of the C=N bond and chemical shifts of neighboring atoms in ^1^H and ^13^C NMR spectra (see Appendix A for details on stereochemistry elucidation) [33,34]. The assigned configuration was additionally confirmed by 2D ^1^H−^1^H NOESY correlations and ^1^*J*_CH_ coupling constants in the C(N)H unit [35] for dioxime **2a**. 

Dioximes **2a–o** were then subjected to catalytic hydrogenation conditions (Figure 2). Two heterogeneous catalysts, namely palladium on charcoal (5%-Pd/C) and Raney^®^ nickel (Ra-Ni), were chosen for this study since these cheap and readily available catalysts often show the best performance in the hydrogenation of oximes [26,36]. It was found that hydrogenation of a model dialdooxime **2a** gave the desired piperazine **1a** in 44% yield with 5%-Pd/C catalyst at 40 bar H_2_/50 °C (method A1), while in the reaction with Ra-Ni (method B1), a complex mixture of products formed. The protection of piperazine **1a** with a Boc group during the hydrogenation was advantageous in terms of the product yield and isolation simplicity (cf. yields of products **1a** and *Boc*-**1a**). Optimized conditions (method A2) were successfully applied to a series of dialdooximes **2a–h** that gave the corresponding Boc-piperazines *Boc*-**1** in moderate to good yields. Expectedly, the alkene moiety in dioxime **2d** and the benzylic C−N bond in dioximes **2e**,**f** was not tolerated under hydrogenation conditions with Pd/C, leading to piperazines *Boc*-**1d’** and *Boc*-**1e’**. The reductive debenzylation process was suppressed by using the Ra-Ni catalyst, as demonstrated by a synthesis of chiral piperazine *Boc*-**1f** from dioxime **2f** derived from α-phenylethylamine by method B2.

Importantly, the developed approach allowed the building up of a piperazine ring in amino acid derivatives, as shown for glycine and L-leucine esters (products *Boc*-**1g** and *Boc*-**1h**) as examples. We believe that piperazine-modified peptides can be prepared in a similar manner.

In contrast to dialdooximes **2a−h**, diketooximes **2i−o** underwent the desired reductive cyclization with Ra-Ni catalyst (Figure 2), while poor conversion was observed with 5%-Pd/C (studies with model substrate **2i**). Moreover, protection with Boc_2_O (method B2) was inefficient in this case due to the small reaction rate attributed to a sterically encumbered environment around the nitrogen atom in 2,6-disubstituted piperazines. Thus, protection with more reactive propionic anhydride (method B3) or no protection (method B1) was performed in the hydrogenation of diketooximes **2i−o**. Using these procedures, a series of 2,6-disubstituted piperazines **1** or their *N*-propionyl derivatives *EtCO*-**1** was successfully prepared (Figure 2). Note that the benzylic C−N bonds remained intact in products *EtCO*-**1k**,**m**, and **1o**.

In the reductive cyclization of diketooximes **2i−o**, *cis*-isomers of the corresponding piperazines **1** formed predominantly. In some cases, the process was stereospecific, and only 2,6-*cis*-isomers were obtained. Note that 2,6-diaryl-substituted piperazines of type **1o** have scarcely been described in the literature, and those which have been reported have the 2,6-*trans*-configuration [18]. So far, the reductive cyclization of dioximes **2** appears to be the only available synthetic route to *cis*-2,6-diaryl-piperazines [37]. The comparison of NMR spectra of the obtained piperazines with those previously reported for *cis*-**1n** [38] and *trans*-**1o** [18] confirmed the stereochemical assignment (see Appendix A for details on stereochemistry elucidation).

In the next stage, we aimed to adapt the developed strategy to access unsymmetrical 2-substituted piperazines (Figure 3a). To accomplish this, a sequential assembly of unsymmetrically substituted dioximes **2** from two different **NSA** precursors was required. However, the reaction of *n*-butylamine with an equimolar amount of **NSA** precursor **3a** gave a complex mixture containing products of mono- and bis-addition (**4a** and **2a**, respectively). Selective Michael addition of one equivalent of nitrosoethylene was accomplished only when a significant excess of *n*-butylamine (12 equiv.) was employed. The monooxime **4a** obtained was then used in a reaction with a second **NSA** precursor **3b**,**d**,**e** that afforded unsymmetrically substituted dioximes **2p−r**. Note that changing the sequence of the addition of **NSA** residues (i.e., initial addition of **3b** to *^n^*BuNH_2_ followed by addition of **3a**) was found to produce dioxime **2p** at a lower overall yield. The reductive cyclization of these dioximes was successfully accomplished by hydrogenation with a 5%-Pd/C catalyst according to method A2 to give the corresponding racemic 2-substituted *N*-Boc-protected piperazines *Boc*-**1p−r** in good yields.

To showcase the utility of the developed strategy, the synthesis of a fused piperazine derivative, namely 1,4-diazabicyclo [4.3.0]nonane **5**, was accomplished (Figure 3b). The titled compound represents a building block in the synthesis of numerous pharmaceutically relevant molecules, in particular, hNK_1_ antagonist Orvepitant [39,40,41,42]. Dioxime **2s** was prepared by the sequential addition of **NSA** precursors **3a** and **3f** to excess benzylamine (the major fraction of benzylamine that did not react in the first stage could be regenerated). In the hydrogenation of dioxime **2s** over 5%-Pd/C (method A1), reductive cyclization of the dioxime motif and catalytic debenzylation occurred. Subsequent lactamization of the transient γ-aminoester upon heating in toluene produced the desired bicyclic product **5**.

The proposed mechanism for the reductive cyclization of dioximes **2** into piperazines **1** is depicted in Figure 4 [26]. The key stages involved catalytic hydrogenolysis of both N−O bonds to give diimine intermediate **I**-**1**, followed by its cyclization to give dihydropyrazine **I**-**2**. Subsequent hydrogenation of the C=N bond in **I-2**, elimination of ammonia, and reduction of dihydropyrazine **I**-**3** afforded the final piperazine product **1**. The observed predominant formation of 2,6-*cis*-isomers of piperazines **1** can be explained by the addition of dihydrogen from the less hindered side of the C=N bond in **I**-**3**, opposite to substituent R^2^. 

## 3. Materials and Methods

Full compound characterization, detailed synthetic procedures, and copies of NMR spectra are provided in Appendix A.

### 3.1. General Information

All the reactions were carried out in oven-dried (150 °C) glassware. NMR spectra were recorded at room temperature with peaks of residual solvents as internal standards. Multiplicities are indicated by s (singlet), d (doublet), t (triplet), q (quartet), m (multiplet), and br (broad). For ^13^C spectra of *Boc-***1c**, *EtCO-***1j**, **2e**, and **4d** apodization with exponential multiplication (3 Hz) was used. ^1^H-^15^N HMBC spectra were recorded using CH_3_NO_2_ as a relative compound. HRMS was measured on an electrospray ionization (ESI) instrument with a time-of-flight (TOF) detector. Column chromatography was performed using silica gel 40–60 μm 60A with petroleum ether−ethyl acetate mixtures as eluents. Analytical thin-layer chromatography was performed on silica gel plates with QF-254. Visualization was accomplished with UV light and/or a solution of ninhydrin/CH_3_CO_2_H in ethanol. 

CH_2_Cl_2_ and Et_3_N were distilled from CaH_2_; Et_2_O was distilled from LiAlH_4_; DMF was distilled from CaH_2_ under reduced pressure. Petroleum ether (PE), methanol, ethanol, CHCl_3_, and ethyl acetate were distilled without drying agents. Raney^®^ nickel (Ra-Ni, ca. 50% slurry in water), 5%-Pd/C, *n*-butylamine, benzylamine, *tert*-butylamine, cyclopentylamine, allylamine, (S)-α-phenylethylamine, propionic anhydride, dmap, glycine ethyl ester hydrochloride, and L-leucine ethyl ester hydrochloride were commercial grade and were used as received. Ene-nitrosoacetals **3a** (R^1^ = H), **3b** (R^1^ = Me), **3c** (R^1^ = Ph), **3d** (R^1^ = Et), **3e** (R^1^ = CH_2_Ph), **3f** (R^1^ = CH_2_CH_2_CO_2_Me) were prepared in one step by the silylation of the corresponding aliphatic nitro compounds, in accordance with literature procedures (see ref. [26] and Appendix A for details).

### 3.2. General Procedures

General procedure for the synthesis of symmetrically substituted dioximes **2a−m,o.** A solution of ene-nitrosoacetal **5** (2.1 mL, 1M in CH_2_Cl_2_) was added dropwise to a solution of an amine (1 mmol) in CH_2_Cl_2_ (1 mL), and the mixture was vigorously stirred at room temperature for 24 h. Then, MeOH (2 mL) was added to the reaction mixture and vigorously stirred for 8 h. Then, the reaction mixture was concentrated under reduced pressure, and the residue was subjected to column chromatography on silica gel.

General procedure for the synthesis of monooximes **4.** Amine (10–14 equiv.) was added to a solution of ene-nitrosoacetal **5** in dichloromethane (1 equiv, 1 M in CH_2_Cl_2_), and the mixture was stirred at room temperature for 24 h. Then, MeOH (5 mL) was added to the mixture and stirred at room temperature for 8 h. The reaction mixture was concentrated under reduced pressure, and the residue was subjected to column chromatography on silica gel (PE:EtOAc = 5:1 → 3:1 → 1:1).

Synthesis of unsymmetrically substituted dioximes **2p−2s**. Ene-nitrosoacetal **5** (0.6 mL, 1M in CH_2_Cl_2_) was added dropwise to a solution of monooxime **4** (0.5 mmol) in CH_2_Cl_2_ (1 mL), and the mixture was vigorously stirred at room temperature for 24 h. Then, MeOH (2 mL) was added to the reaction mixture and vigorously stirred for 8 h. Then, the reaction mixture was concentrated under reduced pressure, and the residue was subjected to column chromatography on silica gel.

General procedure for the synthesis of free piperazines **1** with 5%-Pd/C catalyst (method A1). A 5%-Pd/C catalyst (50 mg per 0.5 mmol of **2**) was added to a solution of dioxime **2** (1 equiv.) in methanol (0.1 M). The vial was placed in a steel autoclave which was flushed and filled with hydrogen to a pressure of ca. 40 bar. Hydrogenation was conducted at this pressure and 50 °C for 6 h with vigorous stirring. Then, the autoclave was cooled to rt and slowly depressurized, the catalyst was filtered off, and the solution was concentrated under reduced pressure. The residue was subjected to column chromatography on silica gel (eluent EtOAc:MeOH = 3:1).

General procedure for the synthesis of Boc-protected piperazines *Boc*-**1** with 5%-Pd/C catalyst (method A2). A 5%-Pd/C catalyst (50 mg per 0.5 mmol of **2**) was added to a solution of dioxime **2** (1 equiv.) and Boc_2_O (3 equiv.) in methanol (0.1 M of **2**). The vial was placed in a steel autoclave which was flushed and filled with hydrogen to a pressure of ca. 40 bar. Hydrogenation was conducted at this pressure and 50 °C for 6 h with vigorous stirring. Then, the autoclave was cooled to rt and slowly depressurized, the catalyst was filtered off, and the solution was concentrated under reduced pressure. The residue was subjected to column chromatography on silica gel (eluent PE:EtOAc = 5:1).

General procedure for the synthesis of free piperazines **1** with Raney nickel catalyst (method B1). A suspension of Ra-Ni (ca. 50 mg per 0.5 mmol of **2**) in methanol (1 mL) was added to a vial containing a solution of dioxime **2** (1 equiv.) in methanol (0.1 M). The vial was placed in a steel autoclave which was flushed and filled with hydrogen to a pressure of ca. 40 bar. Hydrogenation was conducted at this pressure and 50 °C for 6 h with vigorous stirring. Then, the autoclave was cooled to rt and slowly depressurized, the catalyst was filtered off, and the solution was concentrated under reduced pressure. The residue was subjected to column chromatography on silica gel (eluent PE:EtOAc = 5:1 → 3:1 → 1:1 → EtOAc).

General procedure for the synthesis of Boc-protected piperazines *Boc*-**1** with Raney nickel catalyst (method B2). A suspension of Ra-Ni (ca. 50 mg per 0.5 mmol of **2**) in methanol (1 mL) was added to a solution of dioxime **2** (1 equiv.) and Boc_2_O (3 equiv.) in methanol (0.1 M of **2**). The vial was placed in a steel autoclave which was flushed and filled with hydrogen to a pressure of ca. 40 bar. Hydrogenation was conducted at this pressure and 50 °C for 6 h with vigorous stirring. Then, the autoclave was cooled to rt and slowly depressurized, the catalyst was filtered off, and the solution was concentrated under reduced pressure. The residue was subjected to column chromatography on silica gel (eluent PE:EtOAc = 5:1).

General procedure for the synthesis of propionyl-protected piperazines *EtCO*-**1** with Raney nickel catalyst (method B3). A suspension of Ra-Ni (ca. 50 mg per 0.5 mmol of **2**) in methanol (1 mL) was placed into a vial containing a solution of dioxime **2** (1 equiv.) in methanol (0.1 M). The vial was placed in a steel autoclave which was flushed and filled with hydrogen to a pressure of ca. 40 bar. Hydrogenation was conducted at this pressure and 50 °C for 6 h with vigorous stirring. Then, the autoclave was cooled to rt and slowly depressurized, the catalyst was filtered off, and the solution was concentrated under reduced pressure. The residue was dissolved in dichloromethane (3 mL per 0.5 mmol of **2**) and propionic anhydride (3 equiv.), then triethylamine (3 equiv.) and dmap (1 equiv.) were added to the solution. The mixture was placed in a refrigerator (about 0 °C) for 12 h. Then the solution was concentrated under reduced pressure, and the residue was subjected to column chromatography on silica gel (eluent PE:EtOAc = 5:1 → 3:1 → 1:1).

### 3.3. Characterization of Final Products

1-Butylpiperazine (**1a**). The compound was prepared according to the general procedure from dioxime **2a** (90 mg, 0.481 mmol). Yield: 30 mg (44%, method A1). Yield: 0 mg (0%, method B1). Yield: 0 mg (0%, method B2). R_f_ = 0.3 (EtOAc-MeOH, 3:1). Colorless oil. ^1^H NMR (300 MHz, COSY, CDCl_3_) δ 6.76 (br. s, 1H, N*H*), 3.22–3.12 (m, 2H, C*H*_2_), 2.92–2.82 (m, 2H, C*H*_2_), 2.82–2.73 (m, 2H, C*H*_2_), 2.32 (t, J = 8.0 Hz, 2H, C*H*_2_CH_2_CH_2_CH_3_), 2.26–2.12 (m, 2H, C*H*_2_), 1.52–1.37 (m, 2H, CH_2_C*H*_2_CH_2_CH_3_), 1.37–1.23 (m, 2H, CH_2_CH_2_C*H*_2_CH_3_), 0.89 (t, J = 7.3 Hz, 3H, CH_2_CH_2_CH_2_C*H*_3_). ^13^C{^1^H} NMR (75 MHz, DEPT135, CDCl_3_) δ 57.9 (*C*H_2_CH_2_CH_2_CH_3_), 57.5 (*C*H_2_NBu), 51.9 (*C*H_2_NH), 29.2 (CH_2_*C*H_2_CH_2_CH_3_), 20.8 (CH_2_CH_2_*C*H_2_CH_3_), 14.1 (CH_2_CH_2_CH_2_*C*H_3_). ^1^H NMR (300 MHz, DMSO-d_6_) δ 7.96 (br s, 1H, N*H*), 3.04–2.88 (m, 2H, C*H*_2_), 2.85–2.66 (m, 2H, C*H*_2_), 2.49–2.41 (m, 2H, C*H*_2_), 2.26 (t, J = 7.0 Hz, 2H, C*H*_2_CH_2_CH_2_CH_3_), 2.15–1.99 (m, 2H, C*H*_2_), 1.45–1.33 (m, 2H, CH_2_C*H*_2_CH_2_CH_3_), 1.33–1.19 (m, 2H, CH_2_CH_2_C*H*_2_CH_3_), 0.87 (t, J = 7.2 Hz, 3H, CH_2_CH_2_CH_2_C*H*_3_). HRMS (ESI): *m*/*z* calcd. for [C_8_H_19_N_2_]^+^ 143.1543, found 143.1544 [M + H]^+^.

*Tert*-butyl 4-butylpiperazine-1-carboxylate (*Boc*-**1a**). The compound was prepared according to the general procedure (method A2) from dioxime **2a** (90 mg, 0.481 mmol). Yield: 86 mg (74%). R_f_ = 0.8 (PE–EtOAc, 1:1). Colorless oil. ^1^H NMR (300 MHz, CDCl_3_) δ 3.53–3.41 (m, 4H, CH_2_), 2.45–2.39 (m, 4H, CH_2_), 2.35 (t, J = 8.0 Hz, 2H, C*H*_2_CH_2_CH_2_CH_3_), 1.56–1.47 (m, 2H, CH_2_C*H*_2_CH_2_CH_3_), 1.45 (s, 9H, C*H*_3_-C), 1.39–1.25 (m, 2H, CH_2_CH_2_C*H*_2_CH_3_), 0.91 (t, J = 7.3 Hz, 3H, CH_2_CH_2_CH_2_C*H*_3_). ^13^C{^1^H} NMR (75 MHz, DEPT135, CDCl_3_) δ 154.8 (C=O), 79.8 (C), 58.6 (*C*H_2_CH_2_CH_2_CH_3_), 53.1 (*C*H_2_NBu), 42.9 (*C*H_2_NBoc), 28.8 (CH_2_*C*H_2_CH_2_CH_3_), 28.6 (*C*H_3_−C), 20.8 (CH_2_CH_2_*C*H_2_CH_3_), 14.1 (CH_2_CH_2_CH_2_*C*H_3_). HRMS (ESI): *m*/*z* calcd. for [C_13_H_27_N_2_O_2_]^+^ 243.2067, found 243.2066 [M + H]^+^.

*Tert*-butyl 4-(tert-butyl)piperazine-1-carboxylate (*Boc*-**1b**). The compound was prepared according to the general procedure (method A2) from dioxime **2b** (80 mg, 0.428 mmol). Yield: 44 mg (42%). R_f_ = 0.5 (PE-EtOAc, 1:1). White solid. ^1^H NMR (300 MHz, CDCl_3_) δ 3.65–3.18 (m, 4H, C*H*_2_NBoc), 2.68–2.18 (m, 4H, C*H*_2_NC), 1.43 (s, 9H, C*H*_3_CO), 1.04 (s, 9H, C*H*_3_CN). ^13^C{^1^H} NMR (75 MHz, DEPT135, CDCl_3_) δ 154.8 (C=O), 79.5 (C–O), 54.1 (*C*–CH_3_), 45.8 (*C*H_2_NC), 44.2 (*C*H_2_NBoc), 28.6 (*C*H_3_CO), 26.0 (*C*H_3_CN). Mp = 50–53 °C. ^1^H NMR spectrum is in agreement with previously published data [18].

*Tert*-butyl 4-cyclopentylpiperazine-1-carboxylate (*Boc*-**1c**). The compound was prepared according to the general procedure (method A2) from dioxime **2c** (90 mg, 0.452 mmol). Yield: 49 mg (43%). R_f_ = 0.6 (PE-EtOAc, 1:1). Colorless oil. ^1^H NMR (300 MHz, CDCl_3_) δ 3.44 (m, 4H, CH_2_), 2.48 (m, 1H, CH), 2.43 (m, 4H, CH_2_), 1.84 (m, 2H, C*H*_2_CH), 1.76−1.61 (m, 2H, C*H*_2_CH), 1.61−1.48 (m, 2H, C*H*_2_CH_2_CH), 1.45 (s, 9H, CH_3_), 1.40–1.33 (m, 2H, C*H*_2_CH_2_CH). ^13^C{^1^H} NMR (75 MHz, DEPT135, CDCl_3_) δ ^13^C NMR (75 MHz, CDCl_3_) δ 154.9 (C=O), 79.7 (C), 67.6 (CH), 52.2 (*C*H_2_N−Cyp), 43.6 (*C*H_2_NBoc), 30.5 (CH_2_*C*H_2_CH), 28.6 (*C*H_3_−C), 24.2 (*C*H_2_CH_2_CH). ^15^N NMR (300 MHz, HMBC, CDCl_3_) δ −295.8 (*N*-CH), −321.8 (*N*-Boc) (relative to nitromethane). HRMS (ESI): *m*/*z* calcd. for [C_14_H_27_N_2_O_2_]^+^ 255.2067, found 255.2061 [M + H]^+^.

*Tert*-butyl 4-propylpiperazine-1-carboxylate (*Boc*-**1d’**). The compound was prepared according to the general procedure (method A2) from dioxime **2d** (50 mg, 0.292 mmol). Yield: 45 mg (68%). R_f_ = 0.75 (PE-EtOAc, 1:1). Colorless oil. ^1^H NMR spectrum is in agreement with previously published data [43]. 

Di-*tert*-butyl piperazine-1,4-dicarboxylate (*Boc*-**1e’**). The compound was prepared according to the general procedure (method A2) from dioxime **2e** (100 mg, 0.453 mmol). Yield: 75 mg (63%). R_f_ = 0.85 (PE-EtOAc, 1:1). White solid. ^1^H NMR (300 MHz, CDCl_3_) δ 3.35 (s, 8H, CH_2_), 1.43 (s, 18H, CH_3_). ^13^C{^1^H} NMR (75 MHz, DEPT135, CDCl_3_) δ 154.8 (2 C=O), 80.1 (2 C), 43.6 (4 CH_2_), 28.5 (6 CH_3_). HRMS (ESI): *m*/*z* calcd. for [C_14_H_27_N_2_O_4_]^+^ 287.1965, found 287.1958 [M + H]^+^.

*Tert*-butyl (*S*)-4-(1-phenylethyl)piperazine-1-carboxylate (*Boc*-**1f**). The compound was prepared according to the general procedure (method B2) from dioxime **2f** (90 mg, 0.383 mmol). Yield: 53 mg (48%). Colorless oil. R_f_ = 0.8 (PE–EtOAc, 1:1). [α]_D_ = −30.3 (c = 0.25, MeOH, 26 °C). ^1^H NMR spectrum is in agreement with previously published data [18].

*Tert*-butyl 4-(2-ethoxy-2-oxoethyl)piperazine-1-carboxylate (*Boc*-**1g**). The compound was prepared according to the general procedure (method A2) from dioxime **2g** (55 mg, 0.253 mmol). Yield: 45 mg (65%). R_f_ = 0.8 (PE–EtOAc, 1:1). Colorless oil. ^1^H NMR spectrum is in agreement with previously published data [44].

*Tert*-butyl (*S*)-4-(1-ethoxy-4-methyl-1-oxopentan-2-yl)piperazine-1-carboxylate (*Boc*-**1h**). The compound was prepared according to the general procedure (method A2) from dioxime **2h** (100 mg, 0.366 mmol). Yield: 55 mg (46%). R_f_ = 0.7 (PE–EtOAc, 1:1). [α]_D_ = −18.6 (c = 1, MeOH, 26 °C). Colorless oil. ^1^H NMR (300 MHz, HSQC, CDCl_3_) δ 4.17 (q, J = 7.1 Hz, 2H, C*H*_2_CH_3_), 3.43 (m, 4H, C*H*_2_NBoc), 3.31 (m, 1H, C*H*N), 2.63 (m, 4H, *C*H_2_NCH), 1.62 (m, 3H, CHC*H*_2_CH and C*H*CH_3_), 1.45 (s, 9H, C*H*_3_–C), 1.28 (t, J = 7.1 Hz, 3H, CH_2_C*H*_3_), 0.92 (d, J = 6.2 Hz, 3H, C*H*_3_CH), 0.90 (d, J = 6.2 Hz, 3H, C*H*_3_CH). ^13^C{^1^H} NMR (75 MHz, HSQC, DEPT135, CDCl_3_) δ 171.9 (O–*C*=O), 154.7 (N–*C*=O), 79.6 (*C*), 65.6 (*C*HN), 60.3 (*C*H_2_CH_3_), 49.1 (*C*H_2_NCH), 44.1 (*C*H_2_NBoc), 37.9 (CH*C*H_2_CH), 28.4 (*C*H_3_–C), 25.0 (CH_3_*C*H), 22.5 (*C*H_3_CH), 14.4 (CH_2_*C*H_3_). HRMS (ESI): *m*/*z* calcd. for [C_17_H_33_N_2_O_4_]^+^ 329.2435, found 329.2428 [M + H]^+^. 

1-(4-Butyl-2,6-dimethylpiperazin-1-yl)propan-1-one (*EtCO*-**1i**). The compound was prepared according to the general procedure (method B3) from dioxime **2i** (90 mg, 0.420 mmol). A mixture of stereoisomers that were separated by column chromatography. *Cis*:*trans* = 6.1:1. Yield: 50 mg (53%). R_f_ = 0.5 (PE–EtOAc, 1:1). Colorless oil. *Cis*-isomer: ^1^H NMR (300 MHz, COSY, HSQC, CDCl_3_) δ 4.76–3.72 (m, 2H, C*H*CH_3_), 2.67 (d, J = 11.3 Hz, 2H, C*H*_2_CH), 2.33 (q, J = 7.4 Hz, 2H, COC*H*_2_CH_3_), 2.29 (t, J = 6.9 Hz, 2H, C*H*_2_CH_2_CH_2_CH_3_), 2.04 (dd, J = 11.3, 4.4 Hz, 2H, C*H*_2_CH), 1.52–1.38 (m, 4H, CH_2_CH_2_C*H*_2_CH_3_ and CH_2_C*H*_2_CH_2_CH_3_), 1.33 (d, J = 6.8 Hz, 6H, C*H*_3_CH), 1.15 (t, J = 7.4 Hz, 3H, COCH_2_C*H*_3_), 0.92 (t, J = 7.1 Hz, 3H, CH_2_CH_2_CH_2_C*H*_3_). ^13^C{^1^H} NMR (75 MHz, HSQC, DEPT135, CDCl_3_) δ 172.7 (C=O), 58.2 (*C*H_2_CH), 57.9 (*C*H_2_CH_2_CH_2_CH_3_), 49.1 (*C*HCH_3_), 45.4 (*C*HCH_3_), 29.1 (CH_2_*C*H_2_CH_2_CH_3_), 26.4 (CO*C*H_2_CH_3_), 21.4 (*C*H_3_CH), 21.1 (*C*H_3_CH), 20.5 (CH_2_CH_2_*C*H_2_CH_3_), 14.1 (CH_2_CH_2_CH_2_*C*H_3_), 9.8 (*C*H_3_CH_2_CO). HRMS (ESI): m/z calcd. for [C_13_H_27_N_2_O]^+^ 227.2118, found 227.2109 [M + H]^+^. *Trans*-isomer: ^1^H NMR (300 MHz, COSY, HSQC, CDCl_3_) δ 3.91 (m, 2H, C*H*CH_3_), 2.67 (d, J = 11.0 Hz, 2H, C*H*_2_CH), 2.50–2.20 (m, 6H, C*H*_2_CH, COC*H*_2_CH_3_ and C*H*_2_CH_2_CH_2_CH_3_), 1.52–1.39 (m, 2H, CH_2_C*H*_2_CH_2_CH_3_), 1.36 (d, J = 6.5 Hz, 6H, C*H*_3_CH), 1.32–1.21 (m, 2H, CH_2_CH_2_C*H*_2_CH_3_), 1.13 (t, J = 7.4 Hz, 3H, COCH_2_C*H*_3_), 0.91 (t, J = 7.2 Hz, 3H, CH_2_CH_2_CH_2_C*H*_3_). ^13^C{^1^H} NMR (75 MHz, HSQC, DEPT135, CDCl_3_) δ 175.7 (C=O), 58.5 (*C*H_2_CH_2_CH_2_CH_3_), 58.0 (*C*H_2_CH), 49.1 (*C*HCH_3_), 29.4 (CH_2_*C*H_2_CH_2_CH_3_), 27.9 (CO*C*H_2_CH_3_), 20.6 (CH_2_CH_2_*C*H_2_CH_3_), 20.1 (*C*H_3_CH), 14.2 (CH_2_CH_2_CH_2_*C*H_3_), 9.9 (*C*H_3_CH_2_CO). HRMS (ESI): *m*/*z* calcd. for [C_13_H_27_N_2_O]^+^ 227.2118, found 227.2113 [M + H]^+^. 

1-(4-Cyclopentyl-2,6-dimethylpiperazin-1-yl)propan-1-one (*EtCO*-**1j**). The compound was prepared according to a general procedure (method B3) from dioxime **2j** (90 mg, 0.442 mmol). A mixture of stereoisomers that were separated by column chromatography. *Cis*:*trans* = 2.8:1. Yield: 72 mg (69%). R_f_ = 0.6 (PE–EtOAc, 1:1). *Cis*-isomer: Colorless oil. ^1^H NMR (300 MHz, COSY, HSQC, CDCl_3_) δ 4.69–3.85 (m, 2H, C*H*CH_3_), 2.77 (d, J = 11.2 Hz, 2H, NC*H*_2_CHN), 2.47 (ddd, J = 15.4, 8.4, 7.0 Hz, 1H, CH_2_C*H*CH_2_), 2.34 (q, J = 7.3 Hz, 2H, C*H*_2_CH_3_), 2.07 (dd, J = 11.2, 4.3 Hz, 2H, NC*H*_2_CHN), 1.88–1.75 (m, 2H, C*H*_2_CHC*H*_2_), 1.68 (m, 2H, C*H*_2_CH_2_CH), 1.56 (m, 2H, C*H*_2_CH_2_CH), 1.48–1.37 (m, 2H, C*H*_2_CHC*H*_2_), 1.33 (d, J = 6.8 Hz, 6H, C*H*_3_CH), 1.16 (t, J = 7.3 Hz, 3H, CH_2_C*H*_3_). ^13^C{^1^H} NMR (75 MHz, HSQC, DEPT135, CDCl_3_) δ 172.7 (C=O), 66.5 (*C*HN), 57.1 (N*C*H_2_CHN), 48.8 (*C*HCH_3_), 46.1 (*C*HCH_3_), 30.8 (CH_2_*C*H_2_CH), 26.3 (CO*C*H_2_CH_3_), 24.3 (*C*H_2_CH_2_CH), 21.2 (*C*H_3_CH), 9.8 (*C*H_3_CH_2_). HRMS (ESI): m/z calcd. for [C_14_H_27_N_2_O]^+^ 239.2118, found 239.2117 [M + H]^+^. *Trans*-isomer: Colorless oil. ^1^H NMR (300 MHz, HSQC, CDCl_3_) δ 3.88 (m, 2H, C*H*CH_3_), 2.70 (d, J = 11.3 Hz, 2H, NC*H*_2_CHN), 2.58 (ddd, J = 15.7, 8.6, 6.9 Hz, 1H, CH_2_C*H*CH_2_), 2.49–2.23 (m, 4H, NC*H*_2_CHN and C*H*_2_CH_3_), 1.88–1.44 (m, 8H, CH_2_C*H*_2_CH and C*H*_2_CH_2_CH), 1.34 (d, J = 6.5 Hz, 6H, CHC*H*_3_), 1.11 (t, J = 7.4 Hz, 3H, CH_2_C*H*_3_). ^13^C{^1^H} NMR (75 MHz, HSQC, DEPT135, CDCl_3_) δ 175.7 (C=O), 67.2 (CH_2_*C*HCH_2_), 56.8 (N*C*H_2_CHN), 49.1 (*C*HCH_3_), 30.7 (CH_2_*C*H_2_CH), 30.6 (CH_2_*C*H_2_CH), 27.8 (CO*C*H_2_CH_3_), 24.1 (*C*H_2_CH_2_CH), 20.2 (*C*H_3_CH), 9.8 (*C*H_3_CH_2_CO). HRMS (ESI): *m*/*z* calcd. for [C_14_H_27_N_2_O]^+^ 239.2118, found 239.2115 [M + H]^+^.

1-(2,6-Dimethyl-4-((*S*)-1-phenylethyl)piperazin-1-yl)propan-1-one (*EtCO*-**1k**). The compound was prepared according to the general procedure (method B3) from dioxime **2k** (80 mg, 0.302 mmol). An inseparable mixture of stereoisomers. *Cis*:*trans* = 3:1. Yield: 51 mg (62%). R_f_ = 0.7 (PE–EtOAc, 1:1). [α]_D_ = –12.9 (c = 0.07, MeOH, 25 °C). *Cis*-isomer (characterized in mixture with *trans*-isomer): Colorless oil. ^1^H NMR (300 MHz, CDCl_3_) δ 7.45–7.17 (m, 5H, Ph), 4.62–3.78 (m, 2H, C*H*CH_3_), 3.38 (q, J = 6.7 Hz, 1H, C*H*Ph), 2.89 (m, 1H, C*H*_2_CH), 2.59 (m, 1H, C*H*_2_CH), 2.41–2.22 (q, J = 7.5 Hz, 2H, CH_3_C*H*_2_CO), 2.16 (dd, J = 11.1, 4.4 Hz, 1H, C*H*_2_CH), 2.05 (dd, J = 11.4, 4.3 Hz, 1H, C*H*_2_CH), 1.43–1.29 (m, 6H, 2CHC*H*_3_), 1.29 (d, J = 6.8 Hz, 3H, C*H*_3_CHPh), 1.15 (t, J = 7.5 Hz, 3H, C*H*_3_CH_2_CO). ^13^C{^1^H} NMR (75 MHz, DEPT135, CDCl_3_) δ 172.5 (C=O), 144.4 (i-Ph), 128.3 (m-Ph), 127.5 (o-Ph), 127.1 (p-Ph), 64.1 (*C*HPh), 56.1 (*C*H_2_CH), 54.6 (*C*H_2_CH), 49.2 (CH_2_*C*HCH_3_), 46.8 (CH_2_*C*HCH_3_), 26.3 (CO*C*H_2_CH_3_), 21.3 (*C*H_3_CHCH_2_), 21.0 (*C*H_3_CHCH_2_), 19.8 (*C*H_3_CHPh), 9.7 (COCH_2_*C*H_3_). *Trans*-isomer (characterized in mixture with *cis*-isomer): Colorless oil. ^1^H NMR (300 MHz, CDCl_3_) δ 7.45–7.17 (m, 5H, Ph), 3.88 (m, 2H, C*H*CH_3_), 3.55 (q, J = 6.7 Hz, 1H, C*H*Ph), 2.72 (m, 2H, C*H*_2_CH), 2.41–2.22 (q, J = 7.5 Hz, 2H, CH_3_C*H*_2_CO), 2.31 (m, 2H, C*H*_2_CH), 1.43–1.29 (m, 9H, 2 C*H*_3_CH and C*H*_3_CHPh), 1.15 (t, J = 7.5 Hz, 3H, C*H*_3_CH_2_CO). ^13^C{^1^H} NMR (75 MHz, DEPT135, CDCl_3_) δ 175.9 (C=O), 144.3 (i-Ph), 128.4 (m-Ph), 127.5 (o-Ph), 127.0 (p-Ph), 64.3 (*C*HPh), 55.7 (*C*H_2_CH), 54.6 (*C*H_2_CH), 49.3 (CH_2_*C*HCH_3_), 27.9 (CO*C*H_2_CH_3_), 20.2 (*C*H_3_CHCH_2_), 19.9 (*C*H_3_CHCH_2_), 19.5 (*C*H_3_CHPh), 9.7 (COCH_2_*C*H_3_). HRMS (ESI): *m*/*z* calcd. for [C_17_H_27_N_2_O]^+^ 275.2118, found 275.2113 [M + H]^+^.

Ethyl 2-(3,5-dimethyl-4-propionylpiperazin-1-yl)acetate (*EtCO*-**1l**). The compound was prepared according to the general procedure (method B3) from dioxime **2l** (70 mg, 0.285 mmol). An inseparable mixture of stereoisomers. *Cis*:*trans* = 5.1:1. Yield: 50 mg (69%). R_f_ = 0.7 (PE–EtOAc, 1:1). Colorless oil. *Cis*-isomer (characterized in mixture with *trans*-isomer): ^1^H NMR (300 MHz, HSQC, CDCl_3_) δ 4.42–4.10 (m, 2H, C*H*CH_3_), 4.18 (q, J = 7.1 Hz, 2H, COOC*H*_2_CH_3_), 3.22 (s, 2H, C*H*_2_CO_2_Et), 2.73 (d, J = 11.1 Hz, 2H, CHC*H*_2_), 2.40–2.25 (m, 4H, CHC*H*_2_+COC*H*_2_CH_3_), 1.38 (d, J = 6.7 Hz, 6H, CHC*H*_3_), 1.28 (t, J = 7.1 Hz, 3H, COOCH_2_C*H*_3_), 1.16 (t, J = 7.4 Hz, 3H, COCH_2_C*H*_3_). ^13^C{^1^H} NMR (75 MHz, HSQC, DEPT135, CDCl_3_) δ 172.6 (C=O), 170.2 (*C*OOCH_2_CH_3_), 60.4 (COO*C*H_2_CH_3_), 59.4 (*C*H_2_COOEt), 57.6 (*C*H_2_CH), 49.0 (C*H*CH_3_), 46.4 (C*H*CH_3_), 26.2 (CO*C*H_2_CH_3_), 20.9 (*C*H_3_CH), 19.9 (*C*H_3_CH), 14.2 (COOCH_2_C*H*_3_), 9.6 (COCH_2_*C*H_3_). *Trans*-isomer (characterized in mixture with *trans*-isomer): ^1^H NMR (300 MHz, HSQC, CDCl_3_) δ 4.18 (q, J = 7.1 Hz, 2H, COOC*H*_2_CH_3_), 3.96 (dtd, J = 11.6, 6.5, 3.9 Hz, 2H, C*H*CH_3_), 3.29 (d, J = 7.6 Hz, 2H, C*H*_2_CO_2_Et), 2.92 (dd, J = 11.5, 3.8 Hz, 2H, CHC*H*_2_), 2.57 (dd, J = 11.5, 5.0 Hz, 2H, CHC*H*_2_), 2.40–2.25 (m, 2H, COC*H*_2_CH_3_) 1.38 (d, J = 6.7 Hz, 6H, CHC*H*_3_), 1.28 (t, J = 7.1 Hz, 3H, COOCH_2_C*H*_3_), 1.16 (t, J = 7.4 Hz, 3H, COCH_2_C*H*_3_). ^13^C{^1^H} NMR (75 MHz, HSQC, DEPT135, CDCl_3_) δ 172.6 (C=O), 170.4 (*C*OOCH_2_CH_3_), 60.4 (COO*C*H_2_CH_3_), 59.3 (*C*H_2_COOEt), 56.9 (*C*H_2_CH), 49.0 (C*H*CH_3_), 27.6 (CO*C*H_2_CH_3_), 20.9 (*C*H_3_CH), 14.2 (COOCH_2_C*H*_3_), 9.6 (COCH_2_*C*H_3_). HRMS (ESI): *m*/*z* calcd. for [C_13_H_25_N_2_O_3_]^+^ 257.1860, found 257.1852 [M + H]^+^.

1-(4-Benzyl-2,6-dimethylpiperazin-1-yl)propan-1-one (*EtCO*-**1m**). The compound was prepared according to the general procedure (method B3) from dioxime **2m** (100 mg, 0.4 mmol). Obtained as a sole *cis*-isomer. Yield: 74 mg (71%). R_f_ = 0.6 (PE–EtOAc, 1:1). Colorless oil. ^1^H NMR (300 MHz, CDCl_3_) δ 7.44–7.16 (m, 5H, Ph), 4.56 (m, 1H, C*H*CH_3_), 3.91 (qd, J = 6.4, 4.2 Hz, 1H, C*H*CH_3_), 3.50 (s, 2H, CH_2_Ph), 2.68 (d, J = 11.3 Hz, 2H, C*H*_2_CH), 2.35 (m, 2H, C*H*_2_CH_3_), 2.14 (dd, J = 11.3, 4.2 Hz, 2H, C*H*_2_CH), 1.39 (d, J = 6.4 Hz, 6H, CHC*H*_3_), 1.16 (t, J = 7.5 Hz, 3H, CH_2_C*H*_3_). ^13^C{^1^H} NMR (75 MHz, DEPT135, CDCl_3_) δ 172.6 (C=O), 138.5 (i-Ph), 128.6 (m-Ph), 128.3 (o-Ph), 127.1 (p-Ph), 62.7 (*C*H_2_Ph), 57.9 (*C*H_2_CH), 48.2 (CH_3_*C*H), 44.9 (CH_3_*C*H), 26.3 (CO*C*H_2_CH_3_), 21.3 (*C*H_3_CH), 19.8 (*C*H_3_CH), 9.7 (CH_2_*C*H_3_). HRMS (ESI): *m*/*z* calcd. for [C_16_H_25_N_2_O]^+^ 261.1961, found 261.1958 [M + H]^+^.

*Tert*-butyl (3*S*,5*R*)-3,5-dimethylpiperazine-1-carboxylate (**1n**). The compound was prepared according to the general procedure (method B1) from dioxime **2n** (110 mg, 0.423 mmol). Obtained as a sole *cis*-isomer. Yield: 37 mg (46%). R_f_ = 0.4 (PE–EtOAc, 1:1). Colorless oil. ^1^H NMR spectrum is in agreement with previously published data [38].

(3*S*,5*R*)-1-Butyl-3,5-diphenylpiperazine (**1o**). The compound was prepared according to the general procedure (method B1) from dioxime **2o** (90 mg, 0.296 mmol). Obtained as a sole *cis*-isomer. Yield: 53 mg (61%). R_f_ = 0.3 (PE–EtOAc, 1:1). Colorless oil. ^1^H NMR (300 MHz, HSQC, CDCl_3_) δ 7.62–7.24 (m, 10H, Ph), 4.31–4.19 (m, 2H, C*H*), 3.08 (dd, J = 11.3, 2.6 Hz, 2H, CHC*H*_2_), 2.59–2.45 (m, 2H, C*H*_2_CH_2_CH_2_CH_3_), 2.16 (t, J = 10.8 Hz, 2H, CHC*H*_2_), 1.58 (m, 2H, CH_2_C*H*_2_CH_2_CH_3_), 1.35 (m, 2H, CH_2_CH_2_C*H*_2_CH_3_), 0.93 (t, J = 7.3 Hz, 3H, CH_2_CH_2_CH_2_C*H*_3_) (N–H was not observed). ^13^C{^1^H} NMR (75 MHz, HSQC, DEPT135, CDCl_3_) δ 142.3 (p-Ph), 128.4 (m-Ph), 127.7 (i-Ph), 127.2 (o-Ph), 60.7 (CH*C*H_2_), 59.9 (*C*H), 58.4 (*C*H_2_CH_2_CH_2_CH_3_), 28.4 (CH_2_*C*H_2_CH_2_CH_3_), 20.7 (CH_2_CH_2_*C*H_2_CH_3_), 14.0 (CH_2_CH_2_CH_2_*C*H_3_). HRMS (ESI): *m*/*z* calcd. for [C_20_H_27_N_2_]^+^ 295.2169, found 295.2161 [M + H]^+^.

*Tert*-butyl 4-butyl-2-methylpiperazine-1-carboxylate (*Boc*-**1p**). The compound was prepared according to the general procedure (method A2) from dioxime **2p** (48 mg, 0.239 mmol). Yield: 32 mg (53%). R_f_ = 0.75 (PE–EtOAc, 1:1). Colorless oil. ^1^H NMR (300 MHz, COSY, HSQC, CDCl_3_) δ 4.19 (s, 1H, C*H*), 3.79 (m, 1H, C*H*_2_NBoc), 3.11 (m, 1H, C*H*_2_NBoc), 2.79 (m, 1H, C*H*_2_NBu), 2.67 (m, 1H, *C*H_2_CH), 2.44-2.16 (m, 2H, C*H*_2_CH_2_CH_2_CH_3_), 2.09 (m, 1H, C*H*_2_CH), 1.95 (m, 1H, C*H*_2_NBu), 1.45 (s, 9H, C*H*_3_C), 1.43 (m, 2H, CH_2_C*H*_2_CH_2_CH_3_), 1.28–1.18 (m, 2H, CH_2_CH_2_C*H*_2_CH_3_), 1.14 (d, J = 6.7 Hz, 3H, C*H*_3_CH), 0.81 (t, J = 7.2 Hz, 3H, CH_2_CH_2_CH_2_C*H*_3_). ^13^C{^1^H} NMR (75 MHz, COSY, HSQC, DEPT135, CDCl_3_) δ 150.8 (C=O), 79.4 (C), 58.3 (*C*H_2_CH_2_CH_2_CH_3_), 57.5 (*C*H_2_CH), 53.6 (*C*H_2_NBu), 47.0 (*C*H), 39.0 (*C*H_2_NBoc), 28.9 (CH_2_*C*H_2_CH_2_CH_3_), 28.5 (*C*H_3_−C), 20.6 (CH_2_CH_2_*C*H_2_CH_3_), 16.1 (*C*H_3_CH), 14.0 (CH_2_CH_2_CH_2_CH_3_). HRMS (ESI): m/z calcd. for [C_14_H_29_N_2_O_2_]^+^ 257.2224, found 257.2218 [M + H]^+^. 

*Tert*-butyl 4-butyl-2-ethylpiperazine-1-carboxylate (*Boc*-**1q**). The compound was prepared according to the general procedure (method A2) from dioxime **2q** (50 mg, 0.233 mmol). Yield: 45 mg (71%). R_f_ = 0.75 (PE–EtOAc, 1:1). Colorless oil. ^1^H NMR (300 MHz, COSY, HSQC, CDCl_3_) δ 3.99 (m, 1H, C*H*CH_2_CH_3_), 3.91 (m, 1H, C*H*_2_NBoc), 3.11 (m, 1H, C*H*_2_NBoc), 2.83 (m, 1H, C*H*_2_NBu), 2.49–2.25 (m, 2H, C*H*_2_CH), 2.06 (m, 1H, C*H*_2_NBu), 1.94–1.78 (m, 3H, CH_2_C*H*_2_CH_2_CH_3_ + CHC*H*_2_CH_3_), 1.78–1.62 (m, 3H, CH_2_CH_2_C*H*_2_CH_3_ and CHC*H*_2_CH_3_), 1.46 (s, 9H, C*H*_3_−C), 1.35 (m, 2H, CH_2_CH_2_C*H*_2_CH_3_), 0.94 (t, J = 7.3 Hz, 3H, CH_2_CH_2_CH_2_C*H*_3_), 0.88 (t, J = 7.3 Hz, 3H, CHCH_2_C*H*_3_). ^13^C{^1^H} NMR (75 MHz, COSY, HSQC, DEPT135, CDCl_3_) δ 155.4 (C=O), 79.5 (C), 58.3 (*C*H_2_CH_2_CH_2_CH_3_), 55.0 (*C*H_2_CH), 53.6 (*C*H_2_NBu), 52.6 (*C*HCH_2_CH_3_), 39.0 (*C*H_2_NBoc), 28.4 (*C*H_3_−C), 23.0 (CH_2_*C*H_2_CH_2_CH_3_), 22.1 (CH*C*H_2_CH_3_), 20.5 (CH_2_CH_2_*C*H_2_CH_3_), 14.0 (CH_2_CH_2_CH_2_*C*H_3_), 10.8 (CHCH_2_*C*H_3_). HRMS (ESI): *m*/*z* calcd. for [C_15_H_31_N_2_O_2_]^+^ 271.2380, found 271.2372 [M + H]^+^. 

*Tert*-butyl 2-benzyl-4-butylpiperazine-1-carboxylate (*Boc*-**1r**). The compound was prepared according to the general procedure (method A2) from dioxime **2r** (60 mg, 0.217 mmol). Yield: 54 mg (75%). R_f_ = 0.8 (PE–EtOAc, 1:1). Colorless oil. ^1^H NMR (300 MHz, COSY, HSQC, CDCl_3_) δ 7.45–7.16 (m, 5H, Ph), 4.21 (m, 1H, C*H*), 4.02–3.89 (m, 1H, C*H*_2_NBoc), 3.22 (td, J = 12.9, 3.4 Hz, 1H, C*H*_2_NBoc), 3.10 (dd, J = 13.0, 9.1 Hz, 1H, C*H*_2_Ph), 2.88 (dd, J = 13.0, 6.9 Hz, 1H, C*H*_2_Ph), 2.85 (m, 1H, C*H*_2_NBu), 2.71 (d, J = 11.5 Hz, 1H, C*H*_2_CH), 2.37 (m, 1H, C*H*_2_CH_2_CH_2_CH_3_), 2.22 (m, 1H, C*H*_2_CH_2_CH_2_CH_3_), 2.02 (dt, J = 12.5, 6.2 Hz, 1H, C*H*_2_NBu), 1.95–1.90 (m, 1H, C*H*_2_CH), 1.51–1.43 (m, 4H, CH_2_C*H*_2_CH_2_CH_3_ + CH_2_CH_2_C*H*_2_CH_3_), 1.41 (s, 9H, C*H*_3_−C), 0.95 (t, J = 7.1 Hz, 3H, CH_2_CH_2_CH_2_C*H*_3_). ^13^C{^1^H} NMR (75 MHz, COSY, HSQC, DEPT135, CDCl_3_) δ 154.7 (C=O), 139.5 (i-Ph), 129.5 (m-Ph), 128.4 (o-Ph), 126.1 (p-Ph), 79.5 (C), 58.3 (*C*H_2_CH_2_CH_2_CH_3_), 54.0 (*C*H_2_CH), 53.6 (*C*H_2_NBu), 53.5 (*C*H), 39.3 (*C*H_2_NBoc), 36.2 (*C*H_2_Ph), 28.9 (CH_2_*C*H_2_CH_2_CH_3_), 28.4 (*C*H_3_−C), 20.6 (CH_2_CH_2_*C*H_2_CH_3_), 14.0 (CH_2_CH_2_CH_2_*C*H_3_). HRMS (ESI): *m*/*z* calcd. for [C_20_H_33_N_2_O_2_]^+^ 333.2537, found 333.2528 [M + H]^+^.

Hexahydropyrrolo [1,2-a]pyrazin-6(2*H*)-one (**5**). A 5%-Pd/C catalyst (50 mg) was added to a solution of dioxime **2s** (50 mg, 0.163 mmol) in methanol (2 mL). The vial was placed in a steel autoclave which was flushed and filled with hydrogen to a pressure of ca. 40 bar. Hydrogenation was conducted at this pressure and 50 °C for 5 h with vigorous stirring. Then, the autoclave was cooled to rt and slowly depressurized, the catalyst was filtered off, and the solution was concentrated under reduced pressure. The residue was dissolved in toluene (2.5 mL). Et_3_N (0.25 mL, 1.8 mmol) was added to the solution and left at 90 °C for 2.5 h with vigorous stirring. The residue was subjected to column chromatography on silica gel (eluent EtOAc:MeOH = 5:1 → 3:1 → 2:1 → 1:1) to give 11 mg (48%) of product **5** as a colorless oil. R_f_ = 0.3 (EtOAc–MeOH, 3:1). ^1^H NMR spectrum is in agreement with previously published data [41].

## 4. Conclusions

In conclusion, a general approach to the synthesis of piperazines bearing substituents at carbon and nitrogen atoms was developed utilizing primary amines and nitrosoalkenes as synthons. The method relies on the sequential double Michael addition of nitrosoalkenes to amines to give bis(oximinoalkyl)amines followed by catalytic reductive cyclization of oxime groups. The reductive cyclization of diketooximes is stereoselective and results in *cis*-isomers of 2,6-disubstituted piperazines. Importantly, the method developed in this work allows straightforward structural modification of bioactive molecules (e.g., α-amino acids) by the conversion of a primary amino group into the piperazine ring. Also, the method is appropriate in the preparation of complex pharmaceutically relevant piperazine-based scaffolds, such as 1,4-diazabicyclo[4.3.0]nonanes.

## Data Availability

The data presented in this study are available in Appendix A.

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
