# Peer review of "Building Up a Piperazine Ring from a Primary Amino Group via Catalytic Reductive Cyclization of Dioximes"

_ijms, 2023, doi:10.3390/ijms241411794_

Round 1
Reviewer 1 Report
In this manuscript, a general approach to the synthesis of piperazines bearing substituents at car bon and nitrogen atoms utilizing primary amines and nitrosoalkenes as synthons was developed. The manuscript is well written, though the authors need to address some concerns before it is considered for publication.
1. Could you please add the general mechanism of your synthesis?
2. Would you be able to compile a comprehensive table that incorporates all the characterization outcomes pertaining to your products?
3. Did you check the purity of all your products?
Author Response
We would like to thank you for a positive evaluation of our manuscript. The answers to your comments are given below:
Reviewer’s comment (#1): Could you please add the general mechanism of your synthesis?
Answer: Thank you for this suggestion. The proposed mechanism (Scheme 4) and its brief discussion were added in the revised manuscript.
Reviewer’s comment (#2): Would you be able to compile a comprehensive table that incorporates all the characterization outcomes pertaining to your products?
Answer: Thank you for this suggestion. Two tables with characteristic NMR chemical shifts for compounds 2 were compiled and added in the Supplementary material (Tables S1 and S2, pp. S48 and S49). These tables are helpful for the stereochemical assignment of these products.
Reviewer’s comment (#3): Did you check the purity of all your products?
Answer: Yes. The identity and purity of all products were confirmed by 1H NMR, 13C NMR, DEPT135, 2D NMR (for selected products) and HRMS. For additional confirmation of purity, 1H NMR spectra with internal standard were taken for several products 1. These measurements showed >95% purity of these samples.
Reviewer 2 Report
Publication of Yevgeny V. Pospelov and Alexey Yu. Sukhorukov "Building a piperazine ring from a primary amino group by catalytic reductive cyclization of dioximes" is an interesting new strategy for the formation of a piperazine heterocyclic system, which is very important for the development of new drug-like molecules.
Generally, the publication is very well written throughout the manuscript, with virtually no typos (except for line 441 in the manuscript and supporting info where 6H should be in italics), which is rare, and with an impressive quality of English. The synthesis part seems reasonable and this method would certainly be a valuable addition to the existing ones.
However, I was quite shocked and disappointed with the structure elucidation part, mainly using NMR spectroscopy techniques, especially the quality of most of the NMR spectra. Some spectra cannot be published in any scientific journal. Especially the 13C NMR spectral data. For instance compounds 1c, 1j, 1p, 1q, 2a, 4a and so on… This is only a peak picking of a baseline, not an actual peaks. There are no NH signal in a 1a, it is a baseline integration.
You should try to obtain a better-quality data. Or you can try to apply some NMR processing techniques to an existent NMR spectra in MestreNova, like zero-filling or apodization with exponential multiplication (I would suggest to use 3 or 5 Hz values at least)
The 15N NMR spectroscopy would be beneficial as you modify nitrogen heterocycles but it is not a major concern from my point of view.
Minor concern: Part of the dept 135 nmr spectra are pressented not in a negative phase for CH2 carbon signals in some of the compounds 2c, 4a and so on, please revise.
Major / critical concerns:
I would like to get some proofs / comments how 1 j-n cis/trans isomers were distinguished?
And the isomer ratios for all of the aldoximes, ketoximes. How?
You provide no explanation or supporting data.
Only the short statement “obtained and characterized as dynamic mixtures of E/Z-isomers” but how did you obtain the ratios? Which is a major isomer and the other ones? Maybe it would be beneficial to have this information for a better understanding of this cyclization mechanism.
Why practically all of the NMR spectra were recorded in CDCl3 (except for 2n d-DMSO)?
NMR spectra in CDCl3 for the oximes is practically the worst-case scenario. As you have seen in the 2n proton spectra you get sharp signals of N-OH.
So, when practically all of the NMR spectra were conducted in CDCl3 you can not perform 1D selective NOESY or 2D selective NOESY NMR spectra as you only get one broad signal, or a signal which is practically flat like a baseline in 1H NMR.
In the aldoximes you could try to overcome this by measuring 1JCH coupling constants of the imine.
The effect of stereochemistry on 1JC-H at the sp2 carbons of oxime, hydrazone, and imine derivatives of aldehydes. ROBERTR. FRASER and MONIQUEBRESSE.Can. J . Chem. 61, 576 (1983).
It really works. http://rmn2d.univ-lille1.fr/rmn2d_en/co/chapitre3_2_4_en.html
From my own experience the differences between J coupling constants of different isomers are really large and the “Heteronuclear 2D J-resolved NMR” experiment really works well. And it is identical to the NOESY data, I’ve got a few publications from the identification of different isomers of aldoximes, which were as a building blocks as well.
All in all the NMR spectral data should have been obtained in d-DMSO. You would have some options in the identification of the ratios of these isomers. And the solubility would be definitely higher, you would not have a situation like in 2a compound where the NMR signals are half the size of CDCL3 residual signal which is indicative of a very poor solubility or concentration of a compound. With these low concentrations in part of the compounds from the publication your integrations are not accurate, non publishable.
For the chemistry part I would suggest minor revision / accept in the present form.
For the structure elucidation part, “Reject”.
Sorry.
Best of luck.
Author Response
We would like to thank you for a positive evaluation of our manuscript and a very thorough and helpful analysis of the spectral part. We hope, after the revision made, the quality of the characterization part has improved. The answers to your comments are given below:
Reviewer’s comment (#1): Some spectra cannot be published in any scientific journal. Especially the 13C NMR spectral data. For instance compounds 1c, 1j, 1p, 1q, 2a, 4a and so on… This is only a peak picking of a baseline, not an actual peaks.
Answer: Thank you for pointing out this issue. We are sorry for a poor quality of some NMR spectra, which was partially due to a broadening of signals in N-acyl piperazine products and the presence of several isomeric forms in dioximes. 13C NMR spectra were retaken for products 1c, 1j, 1p, 1q and 2a. The signal/noise ratio is much higher now and the tertiary carbons are clearly seen now. We double-checked the 13C NMR of 4a, and the signals of all carbon atoms are well-seen in the original spectra. In the 1H NMR of 4a, integration of a very broad NH/OH signal was removed.
Reviewer’s comment (#2): There are no NH signal in a 1a, it is a baseline integration.
Answer: Thank you for pointing out this issue. In the 1H NMR of 1a integration of a very broad NH signal was removed. Also, the spectrum was retaken in DMSO-d6 and the NH signal was clearly seen now. The picture of 1H NMR spectrum in DMSO-d6 and its assignment were included in the revised manuscript/SI.
Reviewer’s comment (#3): You should try to obtain a better-quality data. Or you can try to apply some NMR processing techniques to an existent NMR spectra in MestreNova, like zero-filling or apodization with exponential multiplication (I would suggest to use 3 or 5 Hz values at least).
Answer: Thank you for this suggestion. Spectra were retaken and better-quality data were obtained for 1c, 1j, 1p, 1q, 2a. Also, better resolved spectra of products 1a, 2a, 2j were obtained in DMSO-d6. We tried zero-filling and apodization with exponential multiplication (3 Hz values), which gave somewhat better spectral pictures (compounds Boc-1c, EtCO-1j, 2e, 4d).
Reviewer’s comment (#4): The 15N NMR spectroscopy would be beneficial as you modify nitrogen heterocycles but it is not a major concern from my point of view.
Answer: Thank you for this suggestion. 1H-15N HMBC spectra were obtained for two characteristic products, namely piperazine Boc-1c and dioxime 2a. The 15N chemical shifts are in agreement with the expected ones. This data was included in the revised manuscript/SI.
Reviewer’s comment (#5): Minor concern: Part of the dept 135 nmr spectra are pressented not in a negative phase for CH2 carbon signals in some of the compounds 2c, 4a and so on, please revise.
Answer: Thank you. This was corrected.
Reviewer’s comment (#6): I would like to get some proofs / comments how 1 j-n cis/trans isomers were distinguished?
Answer: Thank you for pointing out this issue. The discussion of stereochemistry elucidation in 2,6-disubstituted piperazines 1 was included in the revised Supporting information (page S44) and a brief comment was added in the revised manuscript text. Please, also see the file attached for a detailed answer.
Reviewer’s comment (#7): And the isomer ratios for all of the aldoximes, ketoximes. How? You provide no explanation or supporting data. Only the short statement “obtained and characterized as dynamic mixtures of E/Z-isomers” but how did you obtain the ratios? Which is a major isomer and the other ones? Maybe it would be beneficial to have this information for a better understanding of this cyclization mechanism.
Answer: Thank you for pointing out this issue. The discussion of assignment of E/Z-configuration and determination of isomer ratio in dioximes 2 was included in the revised Supporting information (page S47) and a brief comment was added in the revised manuscript text. Please, also see the file attached for a detailed answer. The effect of oxime stereochemistry on the reductive cyclization is a complicated issue, which was not specially studied (we were not able to separate individual isomers of dioximes 2). However, based on the proposed mechanism (Scheme 4 in the revised manuscript), it can be supposed that this effect is negligible (all isomers should produce the same diimine intermediate).
Reviewer’s comment (#8): Why practically all of the NMR spectra were recorded in CDCl3 (except for 2n d-DMSO)?
Answer: We agree that for oximes 1H NMR spectra in DMSO-d6 are often of better quality than in CDCl3. However, in the case of dioximes 2 it was expected that the residual signal of solvent and water would overlap with some signals of product isomers (for instance, this was the case for 2a). For this reason, spectra were recorded in CDCl3. Nevertheless, we recorded 1H NMR in DMSO-d6 for 2a and 2j (in addition to 2n). These spectra and the assignment were added in the revised SI.
Reviewer’s comment (#9): So, when practically all of the NMR spectra were conducted in CDCl3 you can not perform 1D selective NOESY or 2D selective NOESY NMR spectra as you only get one broad signal, or a signal which is practically flat like a baseline in 1H NMR.
Answer: Thank you for this helpful suggestion. We performed 2D NOESY NMR for dioximes 2a and 2j. Indeed, for 2a characteristic cross-peaks between NOH and CH hydrogens were observed. This data confirmed our assignment of oxime configuration made on the basis of chemical shift comparison (see answer to comment #7). However, for 2j the NOH hydrogen gave only exchange peaks with the residual water, and thus the configuration of C=N bond could not be deduced. Anyway, 2D NOESY data for both compounds were added in the revised SI.
Reviewer’s comment (#10): In the aldoximes you could try to overcome this by measuring 1JCH coupling constants of the imine. The effect of stereochemistry on 1JC-H at the sp2 carbons of oxime, hydrazone, and imine derivatives of aldehydes. ROBERTR. FRASER and MONIQUEBRESSE.Can. J . Chem. 61, 576 (1983).
Answer: Thank you for this helpful suggestion. For determination of 1JCH coupling constants, we performed 1H-13C HSQC experiment without decoupling on 1H nuclei for dioxime 2a (see copies of NMR spectra section in SI). It was shown that for the E-fragment in the E,E-isomer and E,Z-isomer, the constant 1JCH is 170.8 Hz and 150.6 Hz, respectively. For the Z-fragment in the E,Z-isomer, the constant 1JCH is 186.2 Hz. This correlates with the 1JCH constants for oxime isomers reported in the reference you provided. Also, this assignment is in an agreement with our assignment of oxime configuration made on the basis of chemical shift comparison (see answer to comment #7).

Reviewer 3 Report
The authors report on the synthesis of a piperazine building block from a primary amino group, based on previously established method for the construction of piperidine analogues. This is an interesting study, which deserves to be published after minor revision.
line 30 : electrophiles (instead of electophiles)
line 47 : authors argue that these methods can only be used for the synthesis of C-unsubstituted piperazines. Isn't it possible to make C-subst. piperazines using C-substituted diethanolamines? Perhaps, too difficult, starting materials not available ? should be shortly discussed by the authors
line 73 : what do the authors mean with 'dynamix' mixtures of E/Z isomers?
Scheme 2 : in the construction of the dioximes 2, the reaction is first run for 24h in dichloromethane, followed by 8 hours in methanol. What is the rationale for switching solvent?
Scheme 2 : in the box highlighting 2,6-disubstituted piperazines, method B2 is mentioned. According to the text, mainly method B3 was applied
Scheme 2 : indicate in the scheme that R1 can be H, Me or Ph ; it's now mentioned in the caption, but better in the scheme itself
line 110 : how was the cis-relationship of both substituents proven?
Intermediates 1p, 1q and 1r were isolated as racemic mixtures ? this should be explicitly mentioned in the text
Author Response
We would like to thank you for a positive evaluation of our manuscript. The answers to your comments are given below:
Reviewer’s comment (#1): line 30 : electrophiles (instead of electophiles)
Answer: Thank you. This was corrected.
Reviewer’s comment (#2): authors argue that these methods can only be used for the synthesis of C-unsubstituted piperazines. Isn't it possible to make C-subst. piperazines using C-substituted diethanolamines? Perhaps, too difficult, starting materials not available ? should be shortly discussed by the authors
Answer: In the papers dealing with these methods, only C-unsubstituted diethanolamines were used. This is likely due to a limited availability of the corresponding substituted diethanolamine derivatives and their reduced reactivity in these reactions. Also, the use of unsymmetrically C-substituted diethanolamines may result in regioselectivity issues (competitive formation of 2- and 3-substituted piperazine isomers). A short comment on this was included in the revised introduction: “However, these methods were applied only to the synthesis of C-unsubstituted piperazine derivatives most likely due to a limited availability of the corresponding substituted diethanolamine derivatives and their reduced reactivity in these reactions”.
Reviewer’s comment (#3): what do the authors mean with 'dynamix' mixtures of E/Z isomers?
Answer: The isomeric ratio in dioximes 2 changes with time and depends on the solvent because the isomerization takes place slowly at rt (the isomerization process is dynamic at ambient conditions). This is a common issue for oximes. A comment on this was added in the revised manuscript: “Typically, dioximes 2 were obtained and characterized as mixtures of E/Z-isomers (the isomeric ratio slightly changed with time and depended on the solvent). For dialdooximes 2a-h, E,E- and E,Z-isomers were formed in comparable quantities, while for diketooximes 2i-n E,E-isomer was often predominant.”
Reviewer’s comment (#4): in the construction of the dioximes 2, the reaction is first run for 24h in dichloromethane, followed by 8 hours in methanol. What is the rationale for switching solvent?
Answer: Thank you for pointing out this issue. In the reaction of amines with ene-nitrosoacetals 3, O-silyl ethers of dioximes 2 are initially formed. Addition of methanol is required to perform mild desilylation of these ethers. A short comment on this was added in the revised manuscript text. The use of methanol as a solvent on the first stage (instead of dichloromethane) is problematic because reacts with ene-nitrosoacetal 3.
Reviewer’s comment (#5): in the box highlighting 2,6-disubstituted piperazines, method B2 is mentioned. According to the text, mainly method B3 was applied.
Answer: Thank you for pointing out this issue. Method B2 was corrected to B3 under structures of products EtCO-1i-m in Scheme 2.
Reviewer’s comment (#6): indicate in the scheme that R1 can be H, Me or Ph ; it's now mentioned in the caption, but better in the scheme itself
Answer: Thank you. This was added in Scheme 2.
Reviewer’s comment (#7): how was the cis-relationship of both substituents proven?
Answer: Thank you for pointing out this issue. The discussion of stereochemistry elucidation in 2,6-disubstituted piperazines 1 was included in the revised Supporting information (page S44) and a brief comment was added in the revised manuscript text. Please, also see the file attached for a detailed answer.
Reviewer’s comment (#8): Intermediates 1p, 1q and 1r were isolated as racemic mixtures ? this should be explicitly mentioned in the text.
Answer: Thank you for pointing out this issue. A note on this was added in the revised manuscript text.

Round 2
Reviewer 2 Report
Significantly improved.
I recommend to accept this publication to IJMS.
Best of luck.